# Circulating Cancer-Associated Macrophage-like Cells as a Blood-Based Biomarker of Response to Immune Checkpoint Inhibitors

**DOI:** 10.3390/ijms25073752

**Published:** 2024-03-28

**Authors:** Valentina Magri, Gianluigi De Renzi, Luca Marino, Michela De Meo, Marco Siringo, Alain Gelibter, Roberta Gareri, Chiara Cataldi, Giuseppe Giannini, Daniele Santini, Chiara Nicolazzo, Paola Gazzaniga

**Affiliations:** 1Department of Pathology, Oncology and Radiology, Sapienza University of Rome, 00161 Rome, Italy; marco.siringo@uniroma1.it (M.S.); alain.gelibter@uniroma1.it (A.G.); chiara.cataldi@uniroma1.it (C.C.); daniele.santini@uniroma1.it (D.S.); 2Department of Molecular Medicine, Sapienza University of Rome, 00161 Rome, Italy; gianluigi.derenzi@uniroma1.it (G.D.R.); michela.demeo@uniroma1.it (M.D.M.); giuseppe.giannini@uniroma1.it (G.G.); chiara.nicolazzo@uniroma1.it (C.N.); paola.gazzaniga@uniroma1.it (P.G.); 3Department of Mechanical and Aerospace Engineering, Sapienza University of Rome, 00184 Rome, Italy; luca.marino@uniroma1.it; 4UOC di Oncologia Medica, Ospedale Leopoldo Parodi Delfino, 00034 Colleferro, Italy; roberta.gareri@libero.it

**Keywords:** CellSearch^®^, circulating cancer-associated macrophage-like cells, circulating tumor cells, immunotherapy, non-small-cell lung cancer

## Abstract

Evidence has been provided that circulating cancer-associated macrophage-like cell (CAM-L) numbers increase in response to chemotherapy, with an inverse trend compared to circulating tumor cells (CTCs). In the era of evolving cancer immunotherapy, whether CAM-Ls might have a potential role as predictive biomarkers of response has been unexplored. We evaluated whether a serial blood evaluation of CTC to CAM-L ratio might predict response to immune checkpoint inhibitors in a cohort of non-small-cell lung cancer patients. At baseline, CTCs, CAM-Ls, and the CTC/CAM-L ratio significantly correlate with both progression-free survival (PFS) and overall survival (OS). The baseline CTC/CAM-L ratio was significantly different in early progressors (4.28 ± 3.21) compared to long responders (0.42 ± 0.47) (*p* = 0.001). In patients treated with immune checkpoint inhibitors, a CTC/CAM-L ratio ≤ 0.25 at baseline is associated with better PFS and OS. A baseline CTC/CAM-L ratio ≤ 0.25 is statistically significant to discriminate early progressions from durable response. The results of the present pilot study suggest that CAM-Ls together with CTCs could play an important role in evaluating patients treated with cancer immunotherapy.

## 1. Introduction

Immune checkpoint inhibitors (ICIs) are a class of drugs targeting programmed cell death 1 (PD-1) and programmed death-ligand 1 (PD-L1), thus regulating the immune checkpoints and enhancing the function of the immune system for tumor suppression [1]. In non-small-cell lung cancer (NSCLC), following the first approval of nivolumab in 2014, pembrolizumab, atezolizumab, and durvalumab have been successively introduced, showing remarkable efficacy [2]. Despite their large use in current clinical practice, the search for response biomarkers to ICIs is still challenging. Since the success of immunotherapy is highly dependent on the host’s antitumor immune response, several studies investigated how different immune cell populations affect an individual’s response to ICI treatment [1,3,4]. Although several blood-based biomarkers have been investigated in the last few decades, the prediction of ICI response still currently relies on PD-L1 tumor expression, facing the major criticism inherent to tumor biopsy specimen analysis. In the last decade, an increasing number of studies allowed liquid biopsy to be introduced as a noninvasive approach in precision oncology to better address spatial and temporal cancer heterogeneity through longitudinal monitoring of tumors at different time points, helping inform clinical decision making [5,6,7]. Circulating tumor cells (CTCs), as a ‘liquid biopsy’ from blood, provide important prognostic and predictive information in cancer management [8]. In 2004, the U.S. Food and Drug Administration (FDA) approved the CellSearch^®^ platform for prognostic purposes in metastatic breast, colorectal, and prostate cancers, based on CTC enumeration [9]. More recently, the prognostic significance of CTC isolation using CellSearch^®^ was also confirmed in the setting of advanced NSCLC [10]. The original CellSearch^®^ definition of a CTC requires the presence of an epithelial cell adhesion molecule (EpCAM)+, cytokeratin (CK 8, 18 or 19)+, 4′,6-diamidino-2-phenylindole (DAPI)+, and CD45− and was validated by prospective clinical trials. Nevertheless, a more recent re-evaluation of datasets of archived images from prior studies of CTCs analyzed on the CellSearch^®^ allowed for the identification of several unusual CTC images, shedding light on a previously unexpected heterogeneity of CTC populations [11,12,13]. Accordingly, a re-classification of CellSearch^®^ images was recently performed [14]. In this context, many authors have reported that circulating hybrid cells (CHCs), also defined as double-positive cells (characterized by simultaneously expressing epithelial and leukocyte markers), are frequently detected in the blood of cancer patients and associated with worse prognosis [15]. These atypical events have been suggested to originate from cell fusion events between cancer cells and leukocytes [16]. Conversely, other groups have identified a distinct population of unusual circulating cells that are large polymorphic in shape, often polynuclear, with a diffuse CK cytoplasmic distribution and triple macrophages that have internalized dying tumor cells before reentering circulation (cancer-associated macrophage-like cells, CAM-Ls) [17,18,19]. Although cancer macrophages often present with a tumor-promoting phenotype with impaired phagocytic functions, evidence has been provided about a macrophage-dependent elimination of CTCs [20]. Consistently, numbers of CAM-Ls were reported to change in an inverse relationship to classical CTCs, increasing in numbers when treatment is effective, suggesting that they might be a direct result of antitumor immune activity [21]. Indeed, since treatment-sensitive disease is supposed to undergo phagocytosis more frequently than treatment-resistant disease, one would expect that CAM-Ls might indirectly reflect the burden of cancer cells killed by treatment. It is thus conceivable that a ratio of CTCs to CAM-Ls may reflect the inefficiency of the immune system in tumor control, with lower ratios favoring a tumor-killing state. The aim of the present proof-of-concept study was to evaluate whether the CTC to CAM-L ratio might represent a suitable blood-based predictive biomarker for response to ICIs. For this purpose, we re-evaluated a retrospective set of archived CellSearch^®^ images of CTCs isolated from a group of NSCLC patients enrolled in the Italian cohort of the Nivolumab Expanded program between April 2015 and September 2015 at Policlinico Umberto I of Rome.

## 2. Results

A total of 28 patients had a blood draw both at baseline (T0) and at the time of progression of disease (PD). At 3 months after treatment (T1), blood draws were performed in 13 patients (46%). In the remaining 15 patients, the T1 blood evaluation was not performed (11 patients progressed early, while in 4 patients, blood samples were inadequate for CellSearch^®^ analysis). At 6 months after treatment (T2), blood draws were performed in 11 patients (39%). With a median follow-up of 26 months [95% confidence interval (CI): 17.5–34.4], all 28 patients had undergone disease progression. The median progression-free survival (PFS) was 4 months [95% CI: 0–8.8]. Early progressions were observed in 11/28 patients (35%). Durable responses were observed in 11/28 patients (35%).

### 2.1. Serial Analysis of CTCs and CAM-Ls

CellSearch^®^ images from 28 NSCLC patients who had been previously monitored for CTC enumeration were available for CAM-L analyses. At T0, 28 patients (100%) were CTC- and CAM-L-positive (≥1), and the CTC to CAM-L ratio was 2.47 ± 3.05. Of note, at T1, 11/13 patients (85%) were CTC-positive (≥1 CTC), while all patients (100%) had at least one CAM-L detected and, accordingly, the CTC to CAM-L ratio at T1 reduced to a lower value, 0.39 ± 0.79. At the next time check, T2, 5/13 patients (45%) were CTC-positive (≥1 CTC), while 11/13 patients (85%) had at least one CAM-L detected, and the CTC to CAM-L ratio at T2 was 1.07 ± 1.08. At PD, 28 patients (100%) were CTC-positive (≥1 CTC), all (100%) with at least one CAM-L detected, and the CTC to CAM-L ratio at PD was 5.54 ± 3.72. All the numerical values are reported in Table 1. Noticeably, the mean number of CTCs increases from T0 to PD, while the number of CAM-Ls and, obviously, the value of the CTC to CAM-L ratio present an inverse behavior, with larger values at T0 and lower at PD, as seen in Figure 1.

A representative image of CAM-Ls is illustrated in Figure 2, where the different features and morphology of CTCs compared to CAM-Ls are illustrated. Specifically, cells greater than or equal to 30 μm in size with more than one 4′,6-diamidino-2-phenylindole+ nucleus (DAPI), an oblong or amorphous shape, and a diffused CK staining pattern were defined as CAM-Ls.

### 2.2. Prognostic Role of CTCs, CAM-Ls, and CTC/CAM-L Ratio

The set of all patients has been divided into two groups according to the thresholds CTCs ≥ 1 and CAM-Ls ≥ 4 (CTC/CAM-L ratio ≤ 0.25). The CTC cut-off value was defined according to the relevant literature on NSCLC [10,22]. While cut-off values for NSCLC differ between investigators, we decided to use CTC ≥ 1 as a cut-off based on previous studies and to maximize the number of patients that were CTC-positive. The threshold for CAM-Ls was evaluated by means of receiver operating characteristic (ROC) analysis, which optimizes the prediction of PFS below or above the median value. In correspondence with CAM-Ls ≥ 4, the area under the curve reaches the maximum AUC = 0.912 (95% CI: 0.798–1), with a significant difference in PFS values between the two groups (Table 2).

The Kaplan survival curves of PFS for the two groups of patients are shown in Figure 3. At T0, CTCs, CAM-Ls, and the CTC/CAM-L ratio significantly correlate with PFS. At T1, the only variable correlated with PFS was CAM-L count, while at T2, CTC and CAM-L counts were both significantly correlated with PFS (CTC: r = −0.66, *p* = 0.03, CAM-L: r = 0.62, *p* = 0.02). At PD, CTCs and the CTC/CAM-L ratio both significantly correlate with PFS (CTCs: r = −0.42, *p* = 0.02; CTCs/CAM-Ls: r = −0.44, *p* = 0.01), as shown in Table 3.

### 2.3. Predictive Role of CTCs, CAM-Ls, and CTC/CAM-L Ratio

PFS was significantly lower in early progressors compared to long responders (PFS values: 2.4 (1.64–3.17) vs. 13 (7.8–18.2), *p* < 0.001). The baseline values of CTCs, CAM-Ls, and CTC/CAM-L ratios for the two groups are significantly different (Table 4): CTCs 5.2 ± 3.80 (early progressors) vs. 1.73 ± 1.27 (long responders) (*p* = 0.024); CAM-Ls 1.4 ± 0.82 (early progressors) vs. 6.91 ± 4.12 (long responders) (*p* < 0.001); CTCs/CAM-Ls 4.28 ± 3.21 (early progressors) vs. 0.42 ± 0.47 (long responders) (*p* = 0.001), (Table 4).

## 3. Discussion

The present study provides insight into the role of CAM-Ls and their relationship with CTCs as a possible combined biomarker to predict the ICI response. The retrospective analysis has been carried out on a limited set of CellSearch^®^ images of CTCs in a group of NSCLC patients. In fact, the increasing evidence that the canonical definition of CTCs does not cover the large spectrum of unusual images captured opens new perspectives on the interpretations and the possible clinical impact of CHCs. In this framework, circulating cancer cells expressing immune proteins have been further subgrouped into small circulating hybrid cells and larger, multinucleated cancer-associated macrophage-like cells [23]. While some authors consider these events as interchangeable, hypothesizing that they might be the result of cellular fusion between tumor cells and leukocytes, others suggested that they might have a different biological significance, being large multinucleated double-positive cells probably consistent with macrophages that contain phagocytosed tumor debris. Accordingly, CHCs are usually inversely correlated with overall survival [24,25,26]. These cellular events are typically small in size and are characterized by double staining for epithelial markers and CD45 and by a leukocyte-like shape. These cells were observed at different frequencies, with rates ranging from 38% to 86%, and were unrelated to the presence of CTCs. On the contrary, CAM-Ls are larger than CTCs, have multiple nuclei, and contain phagocytosed cancer protein epitopes in the cytoplasm. Studies have suggested that CAM-Ls derive from tumor-associated macrophages (TAMs) that escape the primary tumor and enter the circulation [27]. Differently from CHCs, Adam et al. reported that CAM-L numbers transiently increase in response to chemotherapy, with an inverse trend compared to CTCs, suggesting that they might be the direct result of antitumor immune activity within the tumor, and thus are suitable for measuring cancer cells that are responding to treatment [20]. More specifically, chemotherapy, but not endocrine therapy, was found to be associated with high CAM-L counts, suggesting that the mechanism for CAM-L release into the circulation might be affected by the type of systemic treatment. In the era of evolving cancer immunotherapy, whether CAM-Ls have a potential role as predictive biomarkers of response has been unexplored to date.

Here, for the first time, we provided pilot data concerning the correlation between a CTC/CAM-L ratio ≤ 0.25 and PFS in a group of metastatic NSCLC patients treated with nivolumab in an expanded access program at different time points. We found that CTC/CAM-L ratio ≤ 0.25 at baseline is associated with PFS and OS. Furthermore, we demonstrated that a baseline CTC/CAM-L ratio ≤ 0.25 is statistically significant to discriminate early progressions from durable response. In our cohort, we demonstrated that CTCs and CAM-Ls have an inverse trend, confirming what was previously reported by other authors referring to chemotherapies [28]. While the correlation between high CAM-L counts and response to treatment is more intuitive in courses of chemotherapy, since an increased number of CAM-Ls could mirror the prevalence of a tumor-killing state induced by cytotoxic drugs, this correlation has never been investigated in courses of immunotherapy. Our results are consistent with those of Pore et al. [28], who recently reported a correlation between increasing CAM-L counts, lowered CTC counts, and pathologic complete response (pCR) in breast cancer patients undergoing neoadjuvant chemotherapy (NAC) [28]. The pCR group was more likely to have >10 CAM-Ls post-NAC vs. non-pCR group; furthermore, in a multivariate logistic regression model predicting pCR, the CAM-L count was positively associated with the log-odds of pCR, while CTCs showed a negative trend. 

The present research has some limitations. The analysis is based on a retrospective design, and the sample size evaluated is limited to 28 patients, which shapes the study as a pilot. A further important point in the present results is the absence of the discrimination between macrophage polarization M1 versus M2 in CAM-Ls. This last point is crucial since elucidating the nature of circulating CAM-Ls (M1 vs. M2) could help to explain the inconsistency between our results and other authors concerning their prognostic significance [29]. It is conceivable that, according to the tumor microenvironment and treatment type, some CAM-Ls might reflect an active immune response to cancer cells in circulation (M1), while some others (M2) might help cancer cells to disseminate, with obvious opposite impacts on the patient’s outcome. Our results are consistent with the evidence that anti-PD-1 or PD-L1 immune checkpoint blockade induces M1 macrophage polarization [30]. Although our results lead us to believe that, in our cohort, CAM-Ls are M1 macrophages, the true nature of these cells remains to be determined. In a recent paper, the identification of macrophage subsets was carried out through a machine learning-based approach [31]. The selection of M1 or M2 macrophage phenotypes was accomplished on their cell size and morphology. In the CellSearch^®^ system, a possible answer could be obtained by inserting a CD38 fluorescent-labeled antibody into the fourth channel. In fact, of the four fluorescence channels of CellSearch^®^, three distinguish CTCs from white blood cells, while the empty fourth channel can enable further molecular characterization. Previously, the fourth channel has been extensively used for better phenotyping isolated CTCs, measuring, for example, estrogen receptors (ER), PD-L1, human epidermal growth factor receptor 2 (HER-2), and Ki-67 expression with fluorescent-labeled antibodies [32,33,34]. Most of these specific markers are expressed by tumor cells on their surface as transmembrane receptors, such as HER2 and PD-L1. The staining of these markers is diffused within CAM-Ls and can be helpful in determining the response to anti-HER2 targeted therapy or to immunotherapy, respectively. In different cases, the markers are nuclear receptors, i.e., ER, Ki67.

Our preliminary study analyzed the numerical count of CTCs and CAM-Ls; however, there is evidence that the size of the CAM-L also holds significant prognostic value. In a patient cohort consisting of SCLC and late-stage breast, prostate, and colorectal cancer patients, Tang et al. demonstrated that not only the presence but also the size of CAM-Ls has prognostic value. Particularly, the presence of a single CAM-L ≥ 50 μm is a predictor of worse prognosis compared to the presence of CAM-Ls < 50 μm [35]. A future prospective study will aim to add the evaluation of the dimensional parameter of CAM-Ls in a larger patient cohort.

Establishing a new classification of CellSearch^®^ images and defining algorithms that integrate classical CTCs and unusual cellular events, such as CAM-Ls and CHC, would represent an important future approach in precision oncology, potentially useful in the identification of new biomarkers for more innovative targeted therapies and immunotherapies. Although a point of strength of our study is the adequate power to analyze the prognostic value of CAM-Ls due to the long follow-up time, these preliminary results warrant larger, prospective validation.

## 4. Materials and Methods

### 4.1. Patient Population

A total of 7.5 mL of peripheral blood was previously collected for CTC enumeration from 28 NSCLC patients who were enrolled in the Italian cohort of the Nivolumab Expanded program at Policlinico Umberto I of Rome between April 2015 and September 2015. Briefly, the inclusion criteria were an age of 18 years or older; histologically or cytologically confirmed stage IV squamous NSCLC; disease progression (PD) or recurrence during or after one or two prior systemic treatments for advanced or metastatic disease; recurrent disease within 6 months of completing platinum-based adjuvant, neoadjuvant, or definitive chemoradiation therapy for locally advanced disease. All patients received 3 mg/kg nivolumab, administered intravenously every 2 weeks for up to 24 months or until PD, unacceptable toxicity, or withdrawal of consent. Blood draws for CTC analysis were performed at baseline (T0), 3 months (T1) and 6 months (T2) after the initiation of therapy, and at the time of PD, documented according to the RECIST criteria.

### 4.2. Tumor Response

Early progressors were defined as patients with PD or non-evaluable response due to early death at T1, while long responders were defined as patients who had either SD, PR, or CR, with no progression measured by RECIST v1.1 for at least 6 months.

### 4.3. Enumeration of CAM-Ls and Standard CTCs

For all patients, canonical CTCs, defined as nucleated EpCAM+, CK+, or CD45 events, were enumerated between April and September 2015 through the FDA-approved CellSearch^®^ system (Menarini Silicon Biosystems, Castel Maggiore, Bo, Italy), as previously described [30]. Briefly, 7.5 mL of whole blood was processed by employing the CellSearch^®^ CXC kit. After EpCAM-based immunomagnetic capture, cells were stained with antibodies anti-CK8,18,19-fluorescein isothiocyanate (FITC) and anti-CD45-allophycocyanin (APC) and with DAPI for the detection of the nucleus.

All archived images were further reanalyzed for CAM-Ls and were defined as EpCAM+, CK+, or CD45+ cells, greater than or equal to 30 μm in size, with more than one nucleus, an oblong or amorphous shape, and a diffused CK staining pattern, following the criteria as previously described [28].

### 4.4. Statistical Analysis

The endpoints of the present analysis were PFS and the relationship to the CTC and CAM-L counts. The continuous variables were represented as mean ± standard deviation. The PFS of the patient cohort—expressed in months as median and 95% confidence interval—was evaluated through a Kaplan–Meier analysis. Pearson coefficient was adopted to evaluate the bivariate correlation analysis between the circulating cell count (CTC, CAM-L) and PFS. The results were considered statistically significant with a *p*-value below 0.05. SPSS Statistics software version 26.0 (IBM Corp., Armonk, NY, USA) was used for the statistical analysis.

## Figures and Tables

**Figure 1 ijms-25-03752-f001:**
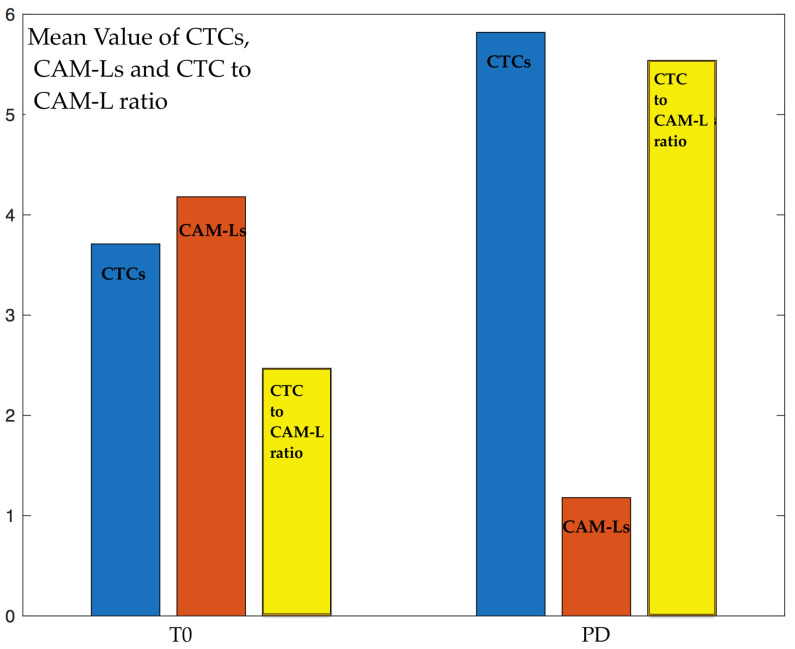
Mean values of CTCs (blue), CAM-Ls (orange), and the CTC to CAM-L ratio (yellow) at baseline T0 and at progression of disease PD.

**Figure 2 ijms-25-03752-f002:**
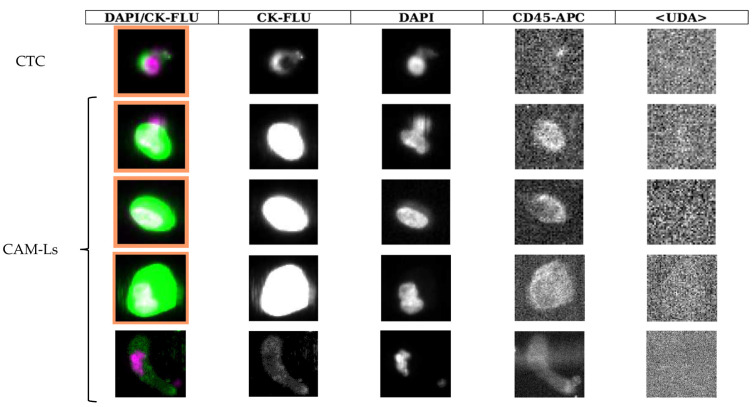
Representative images of cells analyzed using CellSearch^®^ system. One circulating tumor cell (CTC) expresses epithelial marker and is negative for leukocyte marker. Four cancer-associated macrophage-like cells (CAM-Ls) express both epithelial and leukocyte markers. Reading left to right the columns, the first shows a composite image of the cell cytoplasm (green) and nucleus (magenta), and the second to fourth show the gray-scale images of cytokeratin-stained cytoplasm, a DAPI-stained nucleus, and the CD45-APC counterstain, respectively. The last column on the right of the image refers to an additional CellSearch^®^ channel where a phycoerythrin-conjugated user-defined antibody (UDA) can be inserted. CK-FLU: cytokeratin–fluorescein; DAPI: 4′,6-diamidino-2-phenylindole; APC: allophycocyanin.

**Figure 3 ijms-25-03752-f003:**
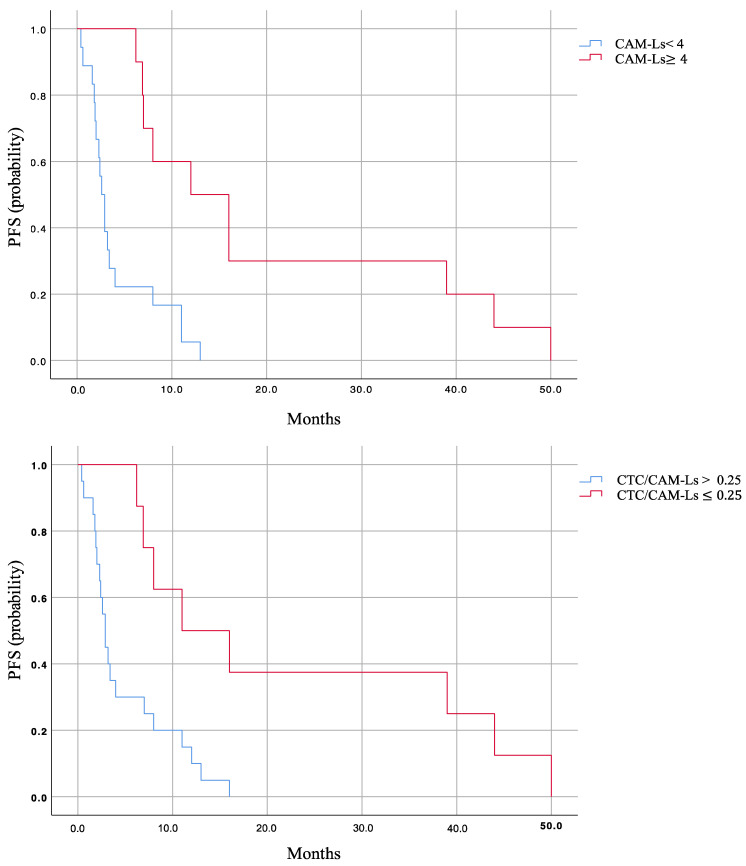
Kaplan–Maier survival curves (probability) for PFS. Comparisons between the CAM-Ls < 4 and CAM-Ls ≥ 4 (**Top**) and between CTCs/CAM-Ls > 0.25 and CTCs/CAM-Ls ≤ 0.25 (**Bottom**); CAM-Ls: cancer-associated macrophage-like cells; CTC: circulating tumor cell; PFS: progression-free survival.

**Table 1 ijms-25-03752-t001:** CTCs, CAM-Ls, and CTC to CAM-L ratio at T0, T1, T2, and PD.

	Baseline (T0) (*n* = 28)	3 Months (T1) (*n* = 13)	6 Months (T2) (*n* = 11)	Progression of Disease (*n* = 28)
CTCs	3.71 ± 3.31	1.53 ± 1.56	0.81 ± 0.98	5.82 ± 4.19
CAM-Ls	4.18 ± 4.21	9.38 ± 5.02	8.81 ± 4.37	1.18 ± 0.46
CTC/CAM-L	2.47 ± 3.05	0.39 ± 0.79	1.07 ± 1.08	5.54 ± 3.72

CAM-Ls: cancer-associated macrophage-like cells; CTCs: circulating tumor cells.

**Table 2 ijms-25-03752-t002:** PFS values (months) for the different CAM-Ls and CTCs/CAM-Ls groups.

	CAM-Ls < 4	CAM-Ls ≥ 4	CTCs/CAM-Ls > 0.25	CTCs/CAM-Ls ≤ 0.25
PFS	2.6 (1.99–3.21)	13 (6.52–19.47)	2.6 (1.91–3.29)	12 (3.73–20.3)
*p*-value	<0.001		<0.001	

CAM-Ls: cancer-associated macrophage-like cells; CTCs: circulating tumor cells; PFS: progression-free survival.

**Table 3 ijms-25-03752-t003:** Pearson correlation coefficients r of CTCs, CAM-Ls, and CTCs/CAM-Ls, at T0, T1, T2, and PD, with the PFS.

	Baseline (T0) (*n* = 28)		3 Months (T1) (*n* = 13)		6 Months (T2) (*n* = 11)		Progression Disease (PD) (*n* = 28)	
	PFS		PFS		PFS		PFS	
	r	*p*	r	*p*	r	*p*	r	*p*
CTCs	−0.43	0.02	−0.38	0.19	−0.66	0.03	−0.42	0.02
CAM-Ls	0.73	<0.001	0.62	0.02	0.75	0.01	0.17	0.37
CTCs/CAM-Ls	−0.47	0.01	−0.27	0.36	−0.5	0.12	−0.44	0.01

CAM-Ls: cancer-associated macrophage-like cells; CTCs: circulating tumor cells; PFS: progression-free survival.

**Table 4 ijms-25-03752-t004:** Baseline values of CTCs, CAM-Ls, CTC/CAM-L ratio, and PFS for all patients, early progressors, and long responders.

	All Patients (*n* = 28)	Early Progressors (*n* = 11)	Long Responders (*n* = 11)	*p*-Value
PFS	4 (0–8.8)	2.4 (1.64–3.17)	13 (7.8–18.2)	<0.001
CTCs (*n* = 28)	3.71 ± 3.31	5.2 ± 3.80	1.73 ± 1.27	0.024
CAM-Ls (*n* = 28)	4.18 ± 4.21	1.4 ± 0.82	6.91 ± 4.12	<0.001
CTC/CAM-L (*n* = 28)	2.47 ± 3.05	4.28 ± 3.21	0.42 ± 0.47	0.001

CAM-Ls: cancer-associated macrophage-like cells; CTCs: circulating tumor cells; PFS: progression-free survival.

## Data Availability

Data will be shared by the corresponding author upon reasonable request.

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
