# Peer review of "Circulating Cancer-Associated Macrophage-like Cells as a Blood-Based Biomarker of Response to Immune Checkpoint Inhibitors"

_ijms, 2024, doi:10.3390/ijms25073752_

Round 1

Reviewer 1 Report

Comments and Suggestions for Authors

The authors of this article have tried to address the challenges of discovering response biomarkers to Immune checkpoint Inhibitors (ICI) treatment. The authors have explored the potential role of circulating Cancer associated macrophage-like cells (CAM-Ls) as predictive biomarkers of immune response. They have shown that CTCs, CAM-Ls, and CTCs/CAM-Ls ratio significantly correlate with both PFS and OS. The findings of this study suggests that CAM-Ls and CTCs could play an important role to evaluate patients treated with cancer immunotherapy. This study shows that CAM-Ls have potential to be used as the prognostic markers to target therapies and immunotherapies. However, there are some instances in the article where authors could have provided more information about some of the methods used.

1.      There a lot of abbreviations used in this article, and I would suggest that the authors to include full forms for each abbreviation, when used for the first time.

2.      Define what T0, T1, T2, and PD is when talking about it in the results section.

3.      In Figure 1, the authors should explain more about the cells, and what is being stained? Why the cell shapes are different in some of the panels. One of the gray panels is not labelled and is not explained in the results section.

4.      Line 122: The authors mention that CTCs cut-off value was defined according to the relevant literature in NSCLC, but there is no reference cited for this.

5.      There are multiple instances where CAM-Ls is misspelled as CAL-Ls.

6.      Line 101: there is no space between in and 13.

7.      The authors talk about the possible inconsistencies due to the nature of CAM-Ls (M1 and M2). Is there a way authors could test M1 and M2 fraction from the CAM-Ls? I was also wondering if authors could incorporate machine learning based methodologies along with the CellSearch platform to distinguish between the M1 and M2 macrophages? There is an article by Hassan M. Rostam et al., in nature scientific reports that used machine learning to differentiate between M1 and M2 macrophages. Herein is the doi for that paper, if the authors are interested in knowing more about it. https://doi.org/10.1038/s41598-017-03780-z. (Image based Machine Learning for identification of macrophage subsets.)

Author Response

Referee 1

The authors of this article have tried to address the challenges of discovering response biomarkers to Immune checkpoint Inhibitors (ICI) treatment. The authors have explored the potential role of circulating Cancer associated macrophage-like cells (CAM-Ls) as predictive biomarkers of immune response. They have shown that CTCs, CAM-Ls, and CTCs/CAM-Ls ratio significantly correlate with both PFS and OS. The findings of this study suggest that CAM-Ls and CTCs could play an important role to evaluate patients treated with cancer immunotherapy. This study shows that CAM-Ls have potential to be used as the prognostic markers to target therapies and immunotherapies. However, there are some instances in the article where authors could have provided more information about some of the methods used.

We thank the reviewer for the useful and valuable comments. Please find herewith the point by point answers to the comments/requests.

  1. There a lot of abbreviations used in this article, and I would suggest that the authors to include full forms for each abbreviation, when used for the first time.

Response: We included full forms for all abbreviations

  1. Define what T0, T1, T2, and PD is when talking about it in the results section.

We defined all the terms as requested

  1. In Figure 1, the authors should explain more about the cells, and what is being stained? Why the cell shapes are different in some of the panels. One of the gray panels is not labelled and is not explained in the results section.

We better explained, in the caption of Figure 1, cells detail regarding staining, shape and meaning of the grey channel.

  1. Line 122: The authors mention that CTCs cut-off value was defined according to the relevant literature in NSCLC, but there is no reference cited for this.

We included the reference

  1. There are multiple instances where CAM-Ls is misspelled as CAM-Ls

We corrected through the text

  1. Line 101: there is no space between in and 13.

We included a space

  1. The authors talk about the possible inconsistencies due to the nature of CAM-Ls (M1 and M2). Is there a way authors could test M1 and M2 fraction from the CAM-Ls? I was also wondering if authors could incorporate machine learning based methodologies along with the CellSearch platform to distinguish between the M1 and M2 macrophages? There is an article by Hassan M. Rostam et al., in nature scientific reports that used machine learning to differentiate between M1 and M2 macrophages. Herein is the doi for that paper, if the authors are interested in knowing more about it. https://doi.org/10.1038/s41

We partially discussed this point in the discussion section. The discussion was extended to consider the paper of  Hassan M. Rostam paper.

Reviewer 2 Report

Comments and Suggestions for Authors

Reviewer comments 

It was my pleasure to review this manuscript (ijms-2901827), because I learnt something new about cancer and the presence of CAM-Ls in the circulation of patients on the therapy. Because the origin and role of CAM-Ls is still unknown in cancer, this manuscript might be very interesting for the scientists  and also for the manufacturer who works on the improvement of these devices. In the present form, the manuscript is not suitable for publishing, and needs major correction. Please, consider my suggestions below for the improvement of your manuscript.

Also some parts of the manuscript for the corrections I have been already marked in your manuscript.

Please write the full name of the abbreviation when it appears for the first time in the text

For example Line 23, 24, 69…..Please, insert full name before NSCLC…. Please, carefully read the manuscript and fix that.

There are technical mistakes of numbers in the main text and tables, please carefully read the manuscript and fix that. Also correct the format of the text.

Results

Please, delete the first two sentences in lines 88 and 89. 

For readers, at the beginning of the results, it is more informative to have the time points when you performed the analyses. For example T0 (baseline), T1 (3 months)..

Some results are repeated, please insert the percentage  from this sentence  ``For all patients we had a baseline blood draw performed (T0), while in13 (46 %) and in 11(39%) we had the T1 and T2 blood draw available, respectively.`` in the sentences in the lines 90-94. 

Also, the authors should escape to repeat the results, for example all results from tables 1 and 2 are already in the text. as well as most of the results from tables 3 and 4. Instead of repeating the numbers it should comment results as a change of obtained values in the selected time points compared to baseline or previous time point. Also in the results add more information about CAM-Ls like their size, evenly the differences in the shape. Maybe it should add some graphical presentation of results instead of table 1, to see the inverse trend of changes in CTCs and CAM-Ls.

The legend of figure 1 is not sufficiently informative, the last column of images does not have any legend. What do these images in the last column represent? Also the material and methods section does not give additional information about use APC, DAPI and PE. 

Please, add more details in the material and methods section, as well as some explanation of figure 1 should be put in the main text. Do figure 1 represent all the CAM-Ls and CTCs you found in the patients or are these representative images of some of them? 

If it is possible, all images in the figure 1 should be the same size (it is more pleasurable for the looking).

Line 104 

Replace 11 patients with 11/13 patients ….. while 13 (100%) with  all patients  had….

The same comment for the lines 106 and 107.

Please, delete sign = in the manuscript , when you present results CTCs=3.71±3.31 in the text.

Line 123

Please insert references after relevant literature in NSCLC. Please, pay attention to how the references should be cited.

Lne 142

Please insert add (r) in the table title. Also in the table add a new row with PFS below the row with baseline, T1…, and replace PFS-r with r and p-value with p. 

Discussion

Please, carefully check used abbreviations in the previous parts of the manuscript. 

What does this abbreviation post-NAC mean? .

The first part of the discussion section is very long and most of the facts are already stated in the introduction section, only the most important facts should  be put here. Please reduce this part and add more discussion about your results as well as correlation of your results with the results of other studies.

Line 201 insert suitable references. 

In the line 225 insert for which type of cells or cellular localization are characteristic specific phenotypes. The 4th channel has been extensively used for better phenotyping isolated CTCs measuring for example ER, PD-L1, BCL-225 2, M30, HER-2, and Ki-67 expression with fluorescent-labeled antibodies [30–32].

Materials i methods 

Line 262

Insert ± between mean and SD.

In M&M section, for the readers who are not familiar with the CellSearch® platform or device, please provide additional information about a principle of distinguishing CMCs cells from other cells.  Describe the number of channels (four) and their usage for the labeling of cells.

Comments on the Quality of English Language

It is ok, only minor correction.

Author Response

Referee 2

We thank the reviewer for the useful and valuable comments. Please find herewith the point by point answers to the comments/requests.

  1. Please write the full name of the abbreviation when it appears for the first time in the text

Response: We included full names for all abbreviations

  1. There are technical mistakes of numbers in the main text and tables, please carefully read the manuscript and fix that. Also correct the format of the text.

Response: OK.

  1. Please, delete the first two sentences in lines 88 and 89.

Response: We removed.

  1. Some results are repeated, please insert the percentage from this sentence  ``For all patients we had a baseline blood draw performed (T0), while in13 (46 %) and in 11(39%) we had the T1 and T2 blood draw available, respectively.`` in the sentences in the lines 90-94.

Response: The percentages were entered as requested, and the sentence in the line 100-102 was deleted.

  1. Also, the authors should escape to repeat the results, for example all results from tables 1 and 2 are already in the text. as well as most of the results from tables 3 and 4. Instead of repeating the numbers it should comment results as a change of obtained values in the selected time points compared to baseline or previous time point. Also in the results add more information about CAM-Ls like their size, evenly the differences in the shape. Maybe it should add some graphical presentation of results instead of table 1, to see the inverse trend of changes in CTCs and CAM-Ls.

Response: OK, modified the text removing some numerical details from the text and leaving in tables 1 and 2. Some details concerning size and shape of CAM-Ls are now introduced in the text. A new figure has been introduced to show the CTCs and CAM-Ls  trend between baseline and progression of disease.

  1. The legend of figure 1 is not sufficiently informative, the last column of images does not have any legend. What do these images in the last column represent? Also the material and methods section does not give additional information about use APC, DAPI and PE.

Response: Ok. We improved the material and methods section as well as the figure 1 caption, as requested.

  1. Please, add more details in the material and methods section, as well as some explanation of figure 1 should be put in the main text. Do figure 1 represent all the CAM-Ls and CTCs you found in the patients or are these representative images of some of them?

Response: Ok. We improved the material and methods section as well as the figure 1 caption and its description in the the text, as requested.

  1. If it is possible, all images in the figure 1 should be the same size (it is more pleasurable for the looking).

Response: Ok. We modified the images as requested.

Line 104

  1. Replace 11 patients with 11/13 patients ….. while 13 (100%) with all patients had….

Response: Ok. We made the changes as requested.

  1. The same comment for the lines 106 and 107.

Response: Ok. We made the changes as requested.

  1. Please, delete sign = in the manuscript, when you present results CTCs=3.71±3.31 in the text.

Response: Ok. We changed the text according to the suggestion.

Line 123 

  1. Please insert references after relevant literature in NSCLC. Please, pay attention to how the references should be cited.

Response: OK, we added the relevant citation.

Line 142

  1. Please insert add (r) in the table title. Also, in the table add a new row with PFS below the row with baseline, T1…, and replace PFS-r with r and p-value with p.

Response: Ok. We modified the Table according to the suggestions

Discussion

  1. Please, carefully check used abbreviations in the previous parts of the manuscript.

Response: OK we checked and updated the manuscript to include all the description of abbreviation adopted in the text.

What does this abbreviation post-NAC mean?

Response: NAC stays for neoadjuvant chemotherapy, now defined in the manuscript.

  1. The first part of the discussion section is very long and most of the facts are already stated in the introduction section, only the most important facts should be put here. Please reduce this part and add more discussion about your results as well as correlation of your results with the results of other studies.

Response: OK we changed the text according to the request.

  1. Line 201 insert suitable references.

Response: OK. We introduced a proper reference.

  1. n the line 225 insert for which type of cells or cellular localization are characteristic specific phenotypes. The 4th channel has been extensively used for better phenotyping isolated CTCs measuring for example ER, PD-L1, BCL-225 2, M30, HER-2, and Ki-67 expression with fluorescent-labeled antibodies [30–32].

Response: ok. We discussed in more details the possible cells or cellular localizations adopted in the 4th channel to a better characterization.

Materials methods

  1. Line 262

Insert ± between mean and SD.

Response: OK, we modified the text.

  1. In M&M section, for the readers who are not familiar with the CellSearch® platform or device, please provide additional information about a principle of distinguishing CMCs cells from other cells. Describe the number of channels (four) and their usage for the labeling of cells.

Response: we provided the additional information, as requested.

Round 2

Reviewer 2 Report

Comments and Suggestions for Authors

Reviewer comments 

Non-muscle invasive bladder cancer or non-small cell lung cancer patients were used in this study. In abstract non-muscle invasive bladder cancer patients in other parts NSCLC patients

Figure 1. Increase the size of letters, or add the type of color  (white, orange and yellow) in the figure legend after CLC.... Add number of cells or CTC/CAM-Ls ratio as a label of y-axis.

Explanation of Figure 1. for CTCs/CAM-Ls ratio in T0 and PD is not correct. 

Author Response

We thank the reviewer for the useful and valuable comments. We modified the text and the Figure 1 according to the indications.

  1. Non-muscle invasive bladder cancer or non-small cell lung cancer patients were used in this study. In abstract non-muscle invasive bladder cancer patients in other parts NSCLC patients

Answer: Ok we modified by introducing non-small cell lung cancer in the text, the previous reference to bladder cancer was a mistake.

  1. Figure 1. Increase the size of letters, or add the type of color (white, orange and yellow) in the figure legend after CLC.... Add number of cells or CTC/CAM-Ls ratio as a label of y-axis.

Answer: Ok we modified by the figure and the corresponding legend.

  1. Explanation of Figure 1. for CTCs/CAM-Ls ratio in T0 and PD is not correct. 

Answer: Ok we modified.
